# On the pattern of interannual polar vortex–ozone co-variability during northern hemispheric winter

Frederik Harzer[1], Hella Garny[2,1], Felix Ploeger[3,4], Harald Bönisch[5], Peter Hoor[6], and Thomas Birner[1,2]

[1]Ludwig-Maximilians-Universität, Meteorological Institute, Munich, Germany
[2]Deutsches Zentrum für Luft- und Raumfahrt, Institute of Atmospheric Physics, Oberpfaffenhofen, Germany
[3]Forschungszentrum Jülich, Institute of Energy and Climate Research, Stratosphere (IEK-7), Jülich, Germany
[4]University of Wuppertal, Institute for Atmospheric and Environmental Research, Wuppertal, Germany
[5]Karlsruhe Institute of Technology, Institute of Meteorology and Climate Research, Karlsruhe, Germany
[6]Johannes Gutenberg University, Institute for Atmospheric Physics, Mainz, Germany

**Correspondence:** Frederik Harzer (frederik.harzer@physik.lmu.de)

**Abstract.** Stratospheric ozone is important for both stratospheric and surface climate. In the lower stratosphere during winter its variability is governed primarily by transport dynamics induced by wave-mean flow interactions. In this work, we analyze interannual co-variations between the distribution of zonal mean ozone and the strength of the polar vortex as a measure of dynamical activity during northern hemispheric winter. Specifically, we study co-variability between the seasonal means of the ozone field from modern reanalyses and polar cap-averaged temperature at $100\,\mathrm{hPa}$, which represents a robust and well-defined index for polar vortex strength. We focus on the vertically resolved structure of the associated extratropical ozone anomalies relative to the winter climatology and shed light on the transport mechanisms that are responsible for this response pattern. In particular, regression analysis in pressure coordinates shows that anomalously weak polar vortex years are associated with three pronounced local ozone maxima just above the polar tropopause, in the lower to mid-stratosphere and near the stratopause. In contrast, in isentropic coordinates, using ERA-Interim reanalysis data only the mid- to lower stratosphere shows increased ozone, while a small negative ozone anomaly appears in the lowermost stratosphere. These differences are related to contributions due to anomalous adiabatic vertical motion, which are implicit in potential temperature coordinates. Our analyses of the ozone budget in the extratropical middle stratosphere show that the polar ozone response maximum around $600\,\mathrm{K}$ and the negative anomalies around $450\,\mathrm{K}$ beneath both reflect the combined effects of anomalous diabatic downwelling and quasi-isentropic eddy mixing, which are associated with consecutive, counteracting anomalous ozone tendencies on daily time scales. We find that approx. $71\,\%$ of the total variability of polar column ozone in the stratosphere is associated with year-by-year variations in polar vortex strength based on ERA5 reanalyses for the winter seasons 1980–2022. MLS observations for 2005–2020 show that around $86\,\%$ can be explained by these co-variations with the polar vortex.

## 1 Introduction

Atmospheric ozone has manifold effects on human health and ecosystems on earth (e. g., Barnes et al., 2022). Furthermore, ozone is known to contribute to climate feedbacks due to its radiative properties (IPCC, 2021; WMO, 2022), emphasizing

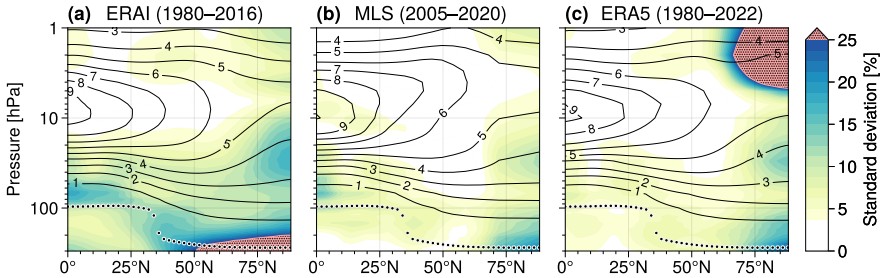

**Figure 1.** Maps of ozone variability (DJF), based on local sample standard deviations relative to the ozone DJF climatology (shown by the solid contour lines in units of ppmv), for ERA-Interim (ERAI), MLS and ERA5. The thick dotted lines show the mean thermal tropopause derived from ERAI data.[1] For ERAI and ERA5, extreme ozone variations (indicated by red shading and stippling) are caused by outliers.

its relevance for atmospheric and climate sciences. For example, Arctic ozone variability was found to substantially impact tropospheric and surface climate (Smith and Polvani, 2014; Calvo et al., 2015; Ivy et al., 2017; Friedel et al., 2022a, b). Diagnostics based on Antarctic ozone depletion have been proposed to improve seasonal forecasts due to robust correlations

with the southern annular mode (Son et al., 2013; Bandoro et al., 2014). Moreover, recent studies reported significant effects of $CO_2$-induced ozone changes on climate sensitivity and on the tropospheric circulation based on models that include interactive ozone (Dietmüller et al., 2014; Nowack et al., 2015, 2018; Chiodo and Polvani, 2017, 2019; Chiodo et al., 2018).

Interannual variability, i. e. variability based on year-by-year anomalies, sheds light on the intrinsic climate fluctuations of the atmosphere and, hence, is essential for understanding long-term trends, e. g., due to external anthropogenic forcings.

Several studies exist in the literature that address the relationship between ozone variability and stratospheric dynamics on this time scale. In particular, planetary waves in the stratosphere, resulting from tropospheric wave activity, have been shown to modulate transport and the zonal-mean distribution of ozone (Hartmann and Garcia, 1979; Garcia and Hartmann, 1980). Based on observational data, a case study by Leovy et al. (1985) demonstrated that the concept of wave breaking can be used to explain co-variability between ozone and potential vorticity in the polar vortex region. Kinnersley and Tung (1998) analyzed

the impact of the quasi-biennial oscillation and planetary wave anomalies on global ozone. Detailed insights on year-by-year variability of atmospheric ozone from both models and observations were provided in subsequent work, mostly based on the upward Eliassen–Palm flux and meridional eddy heat transport at the 100 hPa level as the metrics for stratospheric wave driving (Fusco and Salby, 1999; Randel et al., 2002; Weber et al., 2003; Ma et al., 2004).

While previous studies primarily focused on variations in column ozone, in this work we draw attention to the vertically

resolved pattern of year-by-year ozone variability. To motivate this, consider Fig. 1, which shows maps of local sample standard deviations for zonal mean ozone relative to the winter climatology, based on seasonal-mean data for northern hemispheric winters (December–February, DJF) from two modern reanalyses (ERA-Interim and ERA5) and MLS satellite measurements. More details on the data used are provided in Sect. 2 below. Although these three datasets show considerable quantitative differences, the three maps consistently show a striking tripole structure over the polar cap with isolated variance maxima

---

[1]Throughout this study, thermal tropopause heights are calculated using PyTropD (Adam et al., 2018), following the definition by WMO (1957).

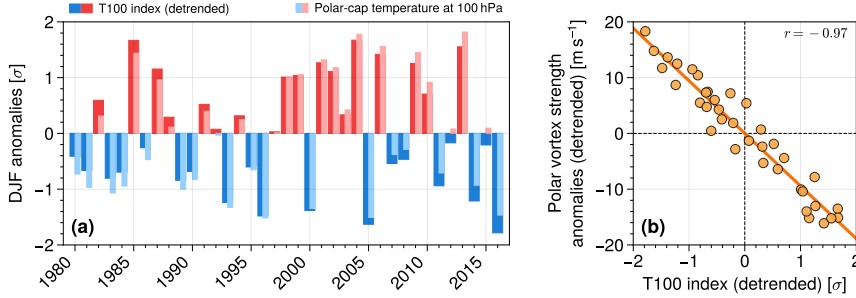

**Figure 2. (a)** T100 index obtained from ERAI reanalysis data, adjusted for a linear trend in time as required for the linear regression approach. The underlying polar-cap averaged temperature anomalies at the 100 hPa level, without this trend removed, are shown by the thin bars. Both time series are standardized by the standard deviation of the T100 index, $\sigma = 2.6$ K. **(b)** Interannual anomalies of the zonal mean zonal wind at the polar vortex around 60°N and 10 hPa, regressed on the T100 index with regression coefficient $b = -9.4\,\mathrm{m\,s^{-1}\,\sigma^{-1}}$ and correlation $r = -0.97$. All results are for ERAI, DJF 1979/80–2015/16.

near the tropopause, in the middle stratosphere and near the stratopause. In this study, we demonstrate that major parts of this variability structure are congruent with interannual co-variations with the stratospheric polar vortex. To do so, we apply a linear regression approach with the seasonal strength of the polar vortex as the predictor, which is characterized by a simple index time series based on polar-cap averaged temperature at 100 hPa. We limit our analyses to the Northern Hemisphere, since the polar vortex features substantially stronger dynamical variability compared to the Southern Hemisphere. This allows for stronger coupling to ozone transport there. We furthermore focus on variations in the polar stratosphere, which are expected to be governed mainly by the Brewer–Dobson circulation, coupled to the variability of polar vortex strength. We find that the resulting winter-mean ozone response pattern may be explained by combined effects of both mixing and residual circulation transport, consistent with previous work focusing on column ozone and the response to sudden stratospheric warmings (e. g., de la Cámara et al., 2018a, b; Hong and Reichler, 2021; Bahramvash Shams et al., 2022).

The paper is structured as follows. In Sect. 2, we list the data used for this study and discuss the setup for the linear regressions. The resulting ozone response patterns are documented in Sect. 3, where we combine regression maps for both pressure and potential temperature coordinates. This allows us to estimate the relative contributions linked to diabatic and adiabatic vertical transport of ozone in the lower polar stratosphere, respectively. In Sect. 4, an ozone budget analysis derived from daily ERA-Interim reanalysis data provides further details on how transport variations produce ozone anomalies at the different levels and regions in the latitude-height plane. Section 5 summarizes our results and provides a brief outlook.

## 2 Data and methods

For the main part of this study, we use model level output of ERA-Interim reanalyses (Dee et al., 2011) for 1979–2016, which we interpolated on pressure and potential temperature levels such that the full vertical resolution of the model was preserved.

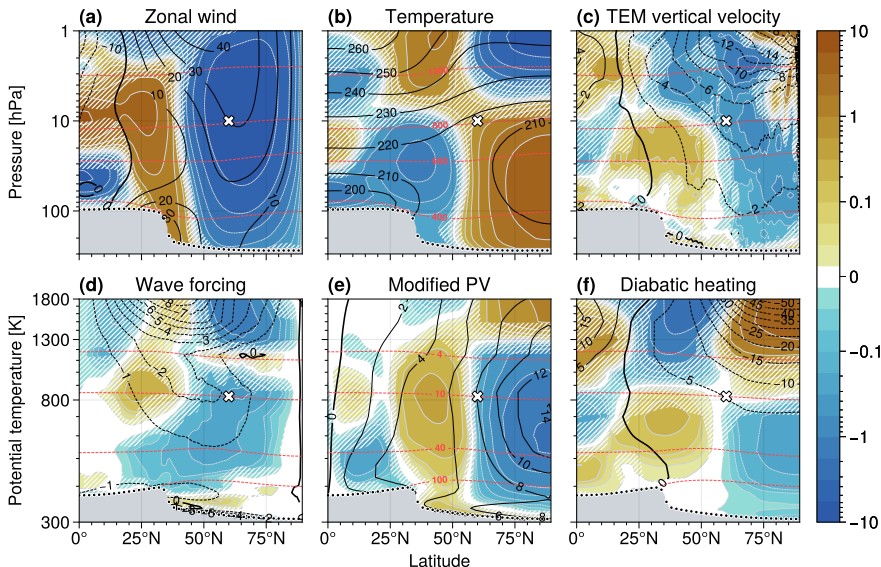

**Figure 3.** Interannual anomalies of the zonal mean **(a)** zonal wind (in units of $\mathrm{m\,s^{-1}}$), **(b)** temperature (K), **(c)** TEM vertical velocity in log-$p$ coordinates ($\mathrm{km\,month^{-1}}$), i. e., $W_{\mathrm{r}} = -(H/p)\,\omega_{\mathrm{r}}$ with scale height $H = 7\,\mathrm{km}$ and $\omega_{\mathrm{r}}$ from eq. (3), **(d)** PV flux $\overline{\hat{v}\sigma\hat{q}}\cos\phi$ (in $\mathrm{m\,s^{-1}\,day^{-1}}$) as a diagnostic for adiabatic wave forcing (Tung, 1986), **(e)** modified PV $\overline{\Pi}^{*} \equiv \overline{q}^{*}\,(\theta/\theta_0)^{-9/2}$ in units of $1\,\mathrm{PVU} = 1 \times 10^{-6}\,\mathrm{K\,m^2\,kg^{-1}\,s^{-1}}$ (Lait, 1994), with density-weighted PV $\overline{q}^{*}$ and $\theta_0 = 350\,\mathrm{K}$, and **(f)** diabatic heating $\overline{Q}^{*}$ ($\mathrm{K\,day^{-1}}$), regressed on the standardized ERAI T100 index. The regression maps are visualized by the color shading and measured per one standard deviation of the index. Note that the panels in the first and second row show the results for pressure and isentropic coordinates, respectively. Hatches indicate where the regressions are not statistically significant. Contour lines show the DJF climatologies of the input fields. The approximate position of the polar vortex around 60°N and 10 hPa is shown by the white markers. The thick dotted lines represent the mean thermal tropopause for DJF. Red dashed lines show selected isentropes (in K, top row) and isobars (in hPa, bottom row), respectively. Note uneven color contour intervals with linear spacing for values within $\pm 0.1$ and logarithmic spacing outside that range. All results are based on ERAI, DJF 1980–2016. Data assimilation and model approximations may cause inconsistencies in the results, which is the case, e. g., for the TEM vertical velocity in **(c)** below the polar stratopause compared to the responses of temperature and diabatic heating in panels **(b)** and **(f)**, respectively.

With that, we consider zonal-mean data on both pressure and isentropic levels on a vertical range roughly from 1000 hPa to
65  1 hPa and 260 K to 1800 K, respectively, with a meridional resolution of about 0.7°. Monthly means are derived from 6-hourly daily data and are combined for seasonally averaged fields for December–February (DJF) 1980–2016. We will refer to the year of January for labeling the corresponding winter season.

Detailed assessments show that ERA-Interim (ERAI) ozone in the lower stratosphere is in reasonable agreement with observations and other reanalysis products (Dragani, 2011; Albers et al., 2018; Davis et al., 2022). We intend to provide further
70  validation by comparing our results obtained from ERAI on pressure levels with those based on the latest ERA5 reanalyses for DJF 1960–2022 with regular 1.5° meridional grid spacing (Hersbach et al., 2019, 2020; Bell et al., 2021). ERA5 data for the pre-satellite era, covering DJF 1960–1980, will be evaluated separately. In addition, we use ozone observations from the

Microwave Limb Sounder (MLS) instrument on board the Aura satellite, data version 5 (Waters et al., 2006; Livesey et al., 2022). The spatial sampling of MLS is comparatively high, with about 3500 profiles along about 15 orbits per day, covering the globe between about 82°S and 82°N. The MLS profile data has been interpolated on potential temperature levels and monthly mean zonal mean climatologies have been compiled for both pressure and potential temperature levels. In this paper, MLS data is considered on pressure levels for DJF 2005–2020 on a non-regular latitudinal grid with mean resolution of approx. 4.2°.

Throughout this study, correlations are measured by local Pearson correlation coefficients $r$ and $p$-values are calculated by a two-sided Wald $t$-test. Statistical significance is assumed if $p \leq 0.05$. 95 % confidence intervals for the correlation coefficients are computed based on Fisher's $z$-transformation.

Co-variability between stratospheric ozone and polar vortex strength is studied by using a simple linear least-squares regression model: first, we define an index time series that is derived from interannual zonal mean temperature, interpolated at the 100 hPa pressure level and averaged over the Northern Hemisphere's polar cap,

$$\overline{T}_{100}(t) \equiv \left\langle \overline{T}\left(t, p = 100\,\mathrm{hPa}, \phi\right) \right\rangle_{\phi \geq 60°\mathrm{N}}, \tag{1}$$

which we will refer to as "T100 index" in the following. Here, overbars indicate the zonal mean and squared brackets denote meridional averaging. The relation between the T100 index and polar vortex strength variability is discussed below within the context of atmospheric dynamics. Then, for a given variable field $\overline{y}$, the corresponding anomalies $\delta\overline{y}$ are regressed on this index according to

$$\delta\overline{y}(t) = a + b \cdot \overline{T}_{100}(t) + \epsilon(t). \tag{2}$$

For both the index and the anomaly time series, potential linear trends across the whole time range under consideration are identified through separate linear least-squares regressions and removed beforehand. Finally, the set of regression coefficients $b(p, \phi)$ yields a "T100 regression map" across the latitude-height plane that quantifies the statistical linear response of $\overline{y}$ if polar stratospheric temperature, measured by the standardized T100 index, is varied by one standard deviation. In this work, this regression model is primarily applied to zonal mean year-by-year anomalies covering DJF winter seasons on the Northern Hemisphere. When daily anomalies are considered, the procedure is similar except that in addition the time series are deseasonalized beforehand.

The detrended and standardized T100 index derived from ERAI reanalyses for DJF 1980–2016 is provided in Fig. 2(a). Choosing this T100 predictor for studying polar vortex co-variability is justified due to its strong dependence on variations of the zonal wind in the polar vortex region at 60°N and 10 hPa, as shown in Fig. 2(b). We therefore obtained a well-defined and powerful proxy diagnostic for the strength of the polar vortex during northern hemispheric winters. Based on temperature as a fundamentally constrained variable instead of zonal wind, this definition features a robust T100 time series across selected other reanalysis products and for moderate changes of the pressure level at which this index is evaluated (not shown).

The strength of the polar vortex and polar-cap lower stratospheric temperature are coupled via downward control (Haynes et al., 1991). Some important variable fields, which are intended to illustrate this mechanism, and their T100 regression maps are provided in Fig. 3. Briefly, starting with anomalously strong wave activity of tropospheric origin, westerly winds in the

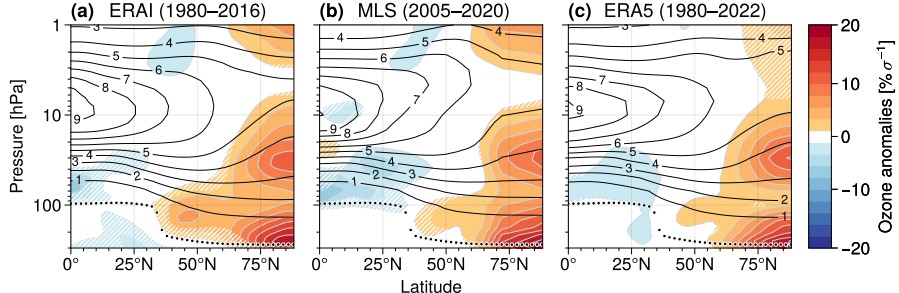

**Figure 4.** Interannual zonal mean ozone on pressure levels from **(a)** ERAI (DJF 1980–2016), **(b)** MLS (2005–2020) and **(c)** ERA5 (1980–2022), regressed on the standardized ERA5 T100 index. The response signatures are provided as relative anomalies (in $\% \, \sigma^{-1}$) based on the respective ozone DJF climatology that is shown by the black contour lines each (in ppmv). The thick dotted lines show the mean thermal tropopause for DJF, derived from ERAI. Other details as in Fig. 3.

polar vortex region are reduced (as shown in Fig. 3a) upon enhanced wave dissipation, inducing a wave-driven poleward residual flow that counteracts the weakening of the background zonal wind due to the Coriolis effect. The wave forcing is indicated by enhanced equatorward potential vorticity (PV) fluxes around the 800 K isentrope in Fig. 3(d), which reduce PV above the polar cap (Fig. 3e). The associated temperature response in Fig. 3(b) is consistent with thermal wind balance.

110 Downwelling (upwelling) is associated with adiabatic warming (cooling), such that by continuity this closes the anomalous residual circulation. This vertical motion (shown with log-$p$ scaling in Fig. 3c) is captured in the Transformed Eulerian Mean (TEM) framework by (e. g., Andrews et al., 1987)

$$\overline{\omega}_{\mathrm{r}} = \overline{\omega} + \frac{1}{a \cos\phi \, \partial_p \overline{\theta}} \partial_\phi \left( \overline{v'\theta'} \cos\phi \right), \tag{3}$$

where $\overline{\omega} = \mathrm{d}p/\mathrm{d}t$ is the vertical velocity in pressure coordinates, $\overline{\theta}$ denotes zonal mean potential temperature, $\overline{v'\theta'}$ indicates

115 the meridional eddy heat transport and $a$ is the earth's radius. The change in potential temperature due to diabatic heating in Fig. 3(f) can be consistently explained to damp the wave-driven temperature perturbations through radiative cooling.

The underlying mechanism has been widely discussed in studies on stratosphere–troposphere coupling, where also the T100 index described above has been first introduced (Baldwin et al., 2019; Domeisen et al., 2020). On sub-seasonal time scales, similar effects are observed, e. g., during sudden stratospheric warmings (Butler et al., 2017; Baldwin et al., 2021).

120 Previous studies often referred to the meridional eddy heat transport at 100 hPa as an indicator for the total available wave driving of the stratosphere (e. g., Fusco and Salby, 1999; Randel et al., 2002; Ma et al., 2004; Weber et al., 2003, 2011; Strahan et al., 2016). However, it is unclear how this wave driving manifests in terms of its latitude-height structure. Furthermore, this metric is more complex than the T100 index, such that it may be less robust across different data products. Following Weber et al. (2011), we used the zonal mean meridional eddy heat flux $\overline{v'T'}$ at 100 hPa averaged between 45°N and 75°N ("VT100

125 index") from ERAI reanalyses, and briefly assessed the resulting signature for ozone co-variability (DJF, 1980–2016). From a regression analysis we found clear similarities in the response patterns obtained for the T100 DJF (discussed below) and

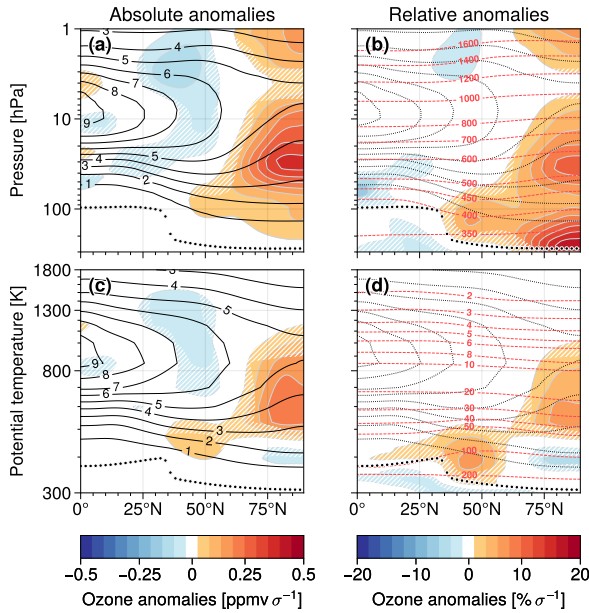

**Figure 5. (a)** T100 regression map for anomalous zonal mean ozone $\overline{\chi}$ on pressure levels (ERAI, DJF 1980–2016). **(b)** Same as in **(a)** but the response signature is given by relative anomalies with respect to the DJF ozone climatology. **(c, d)** Corresponding T100 regression maps for density-weighted zonal mean ozone on potential temperature levels, $\overline{\chi}^* = \overline{\rho_\theta \chi} / \overline{\rho_\theta}$ where $\rho_\theta$ is isentropic density. Details as in Fig. 4.

the VT100 NDJ (November–January, see Appendix A) time series. This time lag between the T100 and VT100 predictors consistently supports the interpretation on anomalous eddy heat fluxes that precede wave-induced deceleration of the polar vortex by several weeks (Newman et al., 2001; Polvani and Waugh, 2004).

## 3 Polar vortex–ozone co-variability: T100 response patterns

In the lower stratosphere, it is mainly the Brewer–Dobson circulation (BDC) that captures ozone transport from the tropics towards higher latitudes (Gettelman et al., 2011; Butchart, 2014). Since wave forcing impacts the strength of both the polar vortex and the BDC, an anomalously weak polar vortex tends to be associated with enhanced poleward ozone transport and, hence, increased polar ozone amounts. Previous studies reported a strong coupling between the strength of the BDC and stratospheric temperature (Fu et al., 2010; Weber et al., 2011; Young et al., 2012), which suggests substantial interannual co-variability between the T100 diagnostic and stratospheric ozone. In this section, we aim to document this T100 ozone response pattern and provide a comparison based on two different perspectives, i. e. using pressure and potential temperature as the vertical coordinate, respectively.

First, Fig. 4 shows zonal mean ozone volume mixing ratios $\overline{\chi}$ on pressure levels, both from ERAI and ERA5 reanalysis data as well as from MLS observations, regressed on the standardized ERA5 T100 index. This index represents an extended

(available for DJF 1980–2022) but essentially equivalent version of the ERAI T100 time series due to its outstanding high correlation, $r > 0.99$. The regression maps are provided in relative units, based on the corresponding ozone DJF climatology, and per one standard deviation of the T100 time series. The stratospheric response signatures show very good agreement among the different datasets in Fig. 4. This is worth noting especially due to the substantial differences in the strength of general ozone variability in Fig. 1. From a more detailed analysis, robust results were found for selected other reanalysis products and for modern chemistry–climate models (not shown; see also von Heydebrand, 2022).

In general, Fig. 4 supports the idea that higher ozone amounts over the polar cap are related to anomalously strong poleward transport due to a stronger BDC during weak polar vortex years. This is consistent with the diabatic heating response in Fig. 3(f) that suggests anomalous upwelling (downwelling) in lower (higher) latitudes for ERAI and, hence, a stronger stratospheric residual circulation. Moreover, the results reflect the vertical structure of year-by-year ozone variability, featuring those three pronounced variance maxima above the polar cap previously discussed with Fig. 1. In addition, the T100 regression maps show another spot of enhanced ozone in the mid-latitudes slightly below the 100 hPa level, which is roughly between the two lowest polar response maxima.

For understanding these features, we can retrieve some first insights from just evaluating the differences among the response signatures obtained for pressure and potential temperature as the vertical coordinate. For that, consider Fig. 5, which shows the T100 regression maps for zonal mean ozone $\overline{\chi}$ on pressure levels (top row) and density-weighted ozone, $\overline{\chi}^* = \overline{\rho_\theta \chi}/\overline{\rho_\theta}$ with isentropic density $\rho_\theta = g^{-1}\partial_\theta p$, for potential temperature coordinates (bottom row). Both computations were obtained from ERAI reanalysis data for DJF 1980–2016. For comparison, the response signatures are displayed not only in relative units (right column) but also as absolute anomalies in parts-per-million (left column), $1\,\text{ppmv} = 1 \times 10^{-6}\,\text{mol}\,\text{mol}^{-1}$.

The results show clear differences among the two coordinate frameworks, which shed light on some relevant transport processes that build up this ozone response signature. For example, consider the two upper polar response maxima around 1 hPa and 30 hPa, respectively, and the corresponding minima at similar altitudes around 40°N more pronounced in the absolute anomalies (top row in Fig. 5). We expect them to arise due to the quadrupole-like response in temperature and TEM vertical velocity (Figs. 3b and 3c), associated with the anomalously weak polar vortex in Fig. 3(a). The sign of the ozone response directly follows from the anomalous up- and downwelling acting on the local vertical gradient of the background ozone distribution. In contrast, this response is much weaker in potential temperature coordinates (bottom row in Fig. 5). This is because vertical ozone transport includes a component due to adiabatic motion, which satisfies $\mathrm{d}\theta/\mathrm{d}t = 0$ and can be thought of shifting iso-surfaces of both ozone and potential temperature simultaneously. Thus for vertical transport, unlike the diabatic part, the adiabatic proportion is implicitly accounted for in isentropic coordinates. Konopka et al. (2009) used similar arguments to explain differences in amplitude of the seasonal cycle of tropical ozone when viewed in pressure and isentropic coordinates.

In the lowermost stratosphere just above the polar tropopause, we find the most remarkable differences among the results for the two coordinate systems used in Fig. 5. On pressure levels, from Fig. 5(b) we find a pronounced ozone response maximum with relative anomalies of more than $15\,\%\,\sigma^{-1}$. Using potential temperature as the vertical coordinate, we instead obtain a decrease of ozone there in Fig. 5(d), suggesting that anomalous downwelling (shown in Fig. 3c) contains a much larger

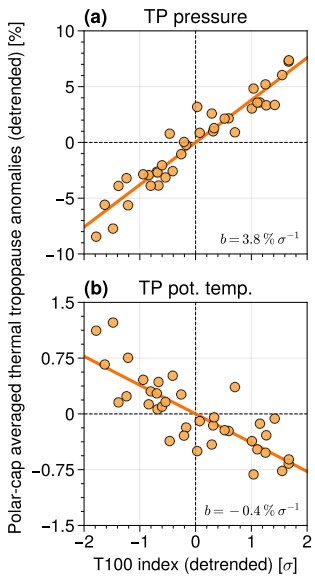

**Figure 6. (a)** Polar-cap averaged thermal tropopause (TP) pressure regressed on the T100 index (ERAI, DJF 1980–2016), with slope $b = 3.8\,\%\,\sigma^{-1}$ from a linear least-squares fit and correlation $r = 0.95$. Much less co-variability is found in **(b)** for polar-cap TP potential temperature ($r = -0.80$).

adiabatic component there compared with higher stratospheric regions. This is consistent with smaller radiative damping rates that are observed in the lower polar stratosphere and that limit the effectiveness of diabatic cooling (Hitchcock et al., 2010).

The large differences between pressure and isentropic coordinates in the lowermost stratosphere are also apparent when considering variations of the polar cap tropopause (Fig. 6): tropopause pressure shows a much larger response to T100 variability (by about a factor of ten) than potential temperature, indicating that polar tropopause height variability is mainly governed by

adiabatic processes. Significantly enhanced ozone then is observed where the substantial lowering of the tropopause height allows for downward transport of stratospheric ozone into former tropospheric regions, which is the case for pressure rather than potential temperature coordinates. As a result, enhanced ozone is found at the polar tropopause in Fig. 5(b), where relative anomalies occur that are large compared to the low ozone amounts usually expected in the upper troposphere.

Taking into account comprehensive research by de la Cámara et al. (2018a, b) on anomalous stratospheric dynamics and

Arctic ozone during sub-seasonal sudden stratospheric warming events, we hypothesize that the remaining T100 response signatures in Figs. 4 and 5 can be explained through variations of the BDC, combining the effects of residual-mean (net mass) downwelling and quasi-isentropic eddy mixing, depending on the local gradients of stratospheric ozone. Within this context, analysing the full ozone budget turned out to be appropriate to investigate the actual roles of these transport contributions.

## 4 Insights on polar vortex–ozone co-variability from the ozone budget

Using isentropic coordinates, the zonal mean ozone budget reads (e. g., Andrews et al., 1987; Plumb, 2002)

$$\partial_t \overline{\chi}^* + \underbrace{\frac{\overline{v}^*}{a} \partial_\phi \overline{\chi}^*}_{-①} + \underbrace{\overline{Q}^* \partial_\theta \overline{\chi}^*}_{-②} = -\overline{\rho_\theta}^{-1} \left[ \underbrace{\frac{1}{a\cos\phi} \partial_\phi \left( \overline{\widehat{v\rho_\theta \chi}} \cos\phi \right)}_{③} + \underbrace{\partial_\theta \overline{\widehat{Q\rho_\theta \chi}}}_{④} \right] + \overline{S}^*. \tag{4}$$

Here, $\chi$ denotes the ozone volume mixing ratio, $Q = d\theta/dt$ is the change in potential temperature due to diabatic heating, $\rho_\theta = g^{-1}\partial_\theta p$ is isentropic density and $S$ represents sources and sinks. We use a density-weighted zonal average, $\overline{\chi}^* \equiv \overline{\rho_\theta \chi}/\overline{\rho_\theta}$ with $\widehat{\chi} = \overline{\chi} - \overline{\chi}^*$ indicating deviations therefrom, and spherical coordinates with geographical latitude $\phi$ and earth's radius

$a$. The tendencies induced by mean flow advection and by eddy mixing are accounted for by the terms ①+② and ③+④, respectively.

Figure 7 provides the T100 regression maps for the relevant dynamical fields and their associated ozone tendencies. We have neglected the contributions due to the mean meridional flow and due to vertical eddy ozone transport, numbered with ① and ④ in eq. (4), since they are small compared to the other processes (not shown). We further omit contributions to ozone

variability due to chemistry, i. e., $S \approx 0$ in eq. (4), which is typically fulfilled in the lower stratosphere, albeit under very cold conditions, ozone depletion may still become important there (e. g., Brasseur and Solomon, 2005). Contributions due to chemistry cannot be neglected in the upper stratosphere; we therefore limit our budget analysis to lower-stratospheric regions, such that $\theta < 900\,\mathrm{K}$ in Fig. 7.

The results show that pronounced anomalous ozone tendencies are mainly found between 400 K and 700 K, where anoma-

lous diabatic vertical transport above the polar cap brings ozone-rich air to lower altitudes, whereas quasi-isentropic eddy mixing counteracts that response by transporting ozone from the polar region to mid-latitudes. Moreover, Figs. 7(d) and 7(e) show that these two leading contributions to ozone variability, ② and ③ as labeled in eq. (4), are largest around 500 K, reflecting the interplay between background ozone gradients and the circulation changes caused by anomalous wave forcing in the polar vortex region according to Fig. 7(c). We furthermore note that, especially in higher latitudes, those anomalous contributions ②

and ③ largely compensate each other. Indeed, the residual ozone tendency is expected to be small, since individual seasonal averages should be in approximate steady state. However, the polar-cap averaged vertical profiles of the responses in Fig. 7(f) show clear differences in the absolute strength of these anomalies, suggesting an imbalance of the seasonal-mean ozone budget instead, especially near 500 K, where the resulting net tendency is negative, $\partial_t \overline{\chi}^* \approx ② + ③ < 0$. This may seem to contradict the enhanced seasonal-mean ozone that is found there during weak polar vortex years (e. g., compare Fig. 5c). However, a

seasonal-mean net negative tendency would merely state that lower ozone values are found at the end compared to the beginning of the season, but would still allow for strong positive ozone anomalies during the season. Moreover, the actual diagnosed seasonal-mean net ozone tendency is in fact slightly positive near 500 K (not shown), which indicates that the detailed quantitative balance of the contributions due to diabatic downwelling and quasi-isentropic eddy mixing is not accurately reproduced in ERAI. This is consistent with Fig. 4, which shows higher ozone anomalies in mid-latitudes in ERAI compared to MLS and

ERA5, presumably due to excessive horizontal mixing.

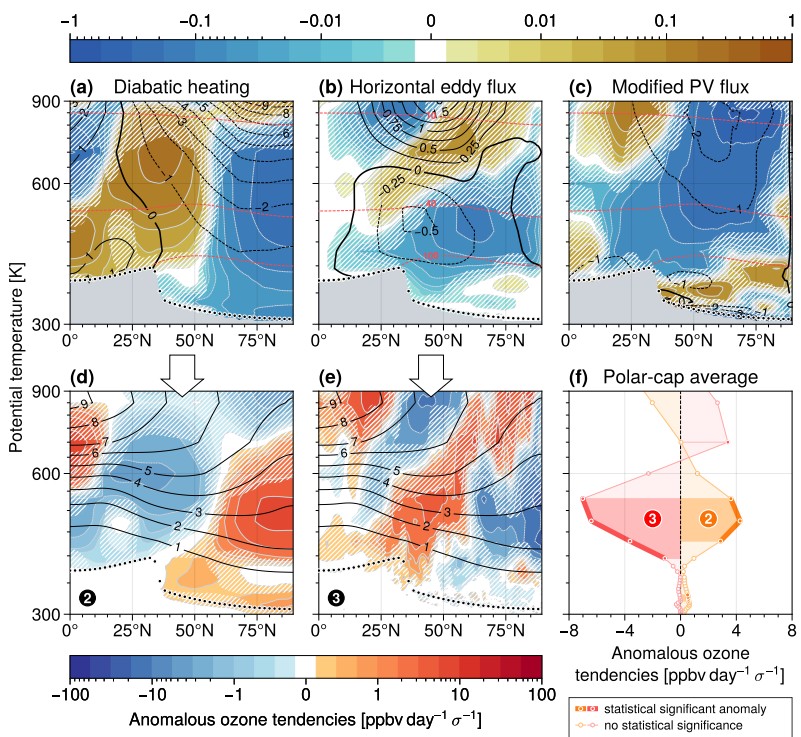

**Figure 7.** (a) Anomalous diabatic heating rates $\overline{Q}^*$ (in K day$^{-1}$) and (b) horizontal eddy ozone transport $\overline{\hat{v}\rho_\theta\hat{\chi}}/\overline{\rho_\theta}$ (in ppmv m s$^{-1}$) regressed on the T100 index (ERAI). (c) The corresponding regression map for the modified PV flux $\overline{\hat{v}\rho_\theta\hat{\Pi}}/\overline{\rho_\theta}$ (in PVU m s$^{-1}$) is added for comparison. Details as in Fig. 3. (d, e) T100 regression maps for the anomalous tendencies ② and ③ from eq. (4). Here, the black contours show the ozone DJF climatology (in ppmv). Note uneven color contour intervals with linear spacing for values within $\pm0.01$ (top row) and $\pm1$ ppbv day$^{-1}\sigma^{-1}$ (bottom row), respectively, and logarithmic spacing outside those ranges. The anomalous ozone tendencies seem to be weak in the lowermost stratosphere and increase strongly for altitudes higher than 400 K due to increased wave forcing above that isentrope as indicated by the anomalous equatorward PV fluxes in (c). Weak positive ozone tendencies due to enhanced diabatic cooling directly above the tropopause in (d) probably arise from an increase in static stability there, e. g., as observed for sudden stratospheric warmings (Grise et al., 2010; Son et al., 2011; Wargan and Coy, 2016), or can be explained as a second-order feedback reacting to changes in the vertical ozone gradients. (f) Polar-cap averaged tendencies, $\langle\partial_t\overline{\chi}^*\rangle_\phi$ for $\phi \geq 60°$N, regressed on the ERAI T100 index. Thick solid lines mark isentropes with statistically significant anomalies. All results are for ERAI, DJF 1980–2016.

These findings need to be treated with caution. In particular, uncertainties in the ERA-Interim reanalyses remain due to data assimilation and due to parameterization of interactive ozone and the associated feedbacks (Davis et al., 2022). Furthermore, diabatic heating rates are solely derived from the model forecasts without any additional constraints and, hence, can introduce additional budget inconsistencies (e. g., Abalos et al., 2015; Monge-Sanz et al., 2022). In conclusion, we do not expect the ozone budget to be closed in ERA-Interim. The cumulative nature and the smaller magnitudes of the winter-mean tendencies may be even more challenging in that respect. We therefore think that such uncertainties only allow for a rather qualitative

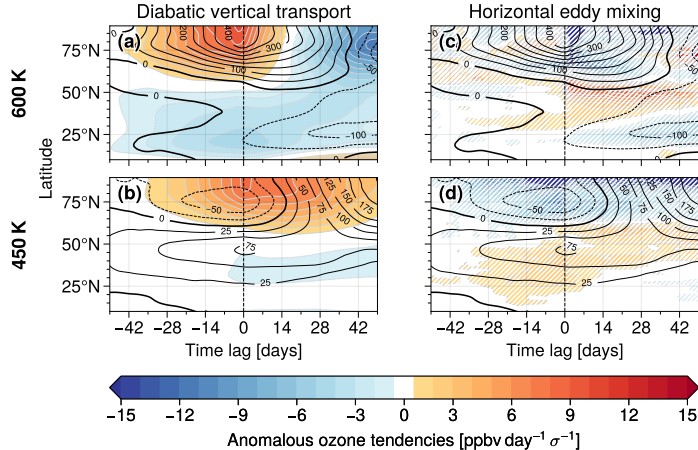

**Figure 8.** Anomalous ozone tendencies associated with **(a, b)** diabatic heating, ② in eq. (4), and **(c, d)** horizontal eddy mixing ③, respectively, regressed on daily T100 anomalies on the 600 K (top row) and 450 K (bottom row) isentrope. Evaluation of a robust positive response at 600 K associated with anomalous diabatic mean flow advection in the tropics (not shown), which occured across all the lag times considered here and possibly includes the response of the quasi-biennial oscillation, remains for subsequent work. Positive (negative) lag times indicate that the field anomalies succeed (precede) the T100 index. Hatching indicates where co-variability is not statistically significant. Black contour lines illustrate T100 lag regressions for daily ozone anomalies $\overline{\chi}^*$ in ppbv per standard deviation $\sigma$ of the daily T100 index. All results are for deseasonalized daily-mean ERAI data, covering December, January and February months in 1979–2016.

estimate of the different ozone transport processes. Additional work is needed to compare these results with other reanalysis products, observational data and model simulations. Limited to the data available from ERA-Interim, we did not find a robust response profile for the resulting net tendency that follows from the anomalous tendencies ②+③ in Fig. 7(f), i.e., statistical

significance (if any) is sensitive to the latitude range chosen for polar-cap averaging (not shown). This suggests that the full seasonal-mean ozone budget may indeed be balanced, such that the T100 response of the net tendency $\partial_t \overline{\chi}^* \approx ②+③$ in eq. (4) vanishes for seasonal averages, assuming that ozone chemistry $\overline{S}^*$ does not play a dominant role in the lower stratosphere.

At this point, these contrasting results presented above complicate reliable conclusions on the underlying transport dynamics based on winter-mean ozone tendencies. Instead, to reveal more insights into the drivers of the seasonal-mean ozone anomalies

it is useful to study daily variability where the relevant transport contributions may not fully compensate, leaving a net ozone tendency that allows to better distinguish cause and effect. To do so, we perform a linear lag-regression analysis using daily averaged fields based on 6-hourly ERAI data for the December, January and February months in 1979–2016. We select two isentropes, 450 K and 600 K, which represent the polar ozone response maximum and the minimum underneath in Fig. 5(c), and assess T100 co-variability for daily ozone and the two relevant dynamical tendencies, ② and ③ in eq. (4). The results for

different lag times, covering 14 weeks in total centered around the T100 anomaly, are shown in Fig. 8. They are consistent with ozone transport taking place, e.g., during sub-seasonal sudden stratospheric warming (SSW) events (de la Cámara et al., 2018a, b; Hong and Reichler, 2021), though here we present lag regressions for the more general case of daily T100 temperature

anomalies, which allows for improved statistics and reduced sensitivity, e. g., on SSW definition thresholds. In Fig. 8, for the 600 K isentrope we find that enhanced polar ozone is mainly induced by anomalous diabatic mean flow downwelling (for negative lag times in Fig. 8a). Subsequently, a reduction of polar ozone takes place due to increased quasi-horizontal eddy mixing in Fig. 8(c) out of the polar cap toward mid-latitudes, which partly arises as a feedback due to strenghtened meridional ozone gradients. Concerning the 450 K level, it turns out that the temporal order of the tendencies involved is now reversed: here, horizontal eddy mixing can be clearly identified to force the negative ozone response in the polar region. This also explains the positive ozone anomaly in the mid-latitudes shown in Figs. 5(b) and (d). Figures 8(a) and 8(b) indicate the anomalies in diabatic mean flow advection propagating downward with time, where they counteract the eddy forcing in lower altitudes. A closer analysis of variations in the underlying wave driving giving rise to anomalous transport, including different contributions due to different parts of the wave spectrum, is beyond the scope of the current analyses and is left for future work.

Overall, our findings here confirm that the complex ozone response signature in the middle stratosphere introduced in Sect. 3 can be explained by the combined effects of horizontal eddy mixing and vertical mean flow advection. We found significant anomalous ozone tendencies as the T100 response on sub-seasonal time scales, which consistently explain the variations in seasonal-mean stratospheric ozone but at the same time seem to not influence ozone net tendency on a seasonal scale. The key for this to happen are those two dominant mechanisms of ozone transport that induce competing response tendencies but occur with some temporal distance.

## 5 Discussion and conclusions

In this paper, we discussed interannual co-variability between the strength of the polar vortex, as indicated by the index time series of polar-cap averaged temperature at 100 hPa, and zonal mean ozone during northern hemispheric winters. We focused on the vertically resolved ozone response pattern in the latitude-height plane, which as far as we know has received only little attention in the literature. In particular, we assessed the intriguing ozone response structure in middle and high latitudes across different altitude levels from two different coordinate perspectives: in pressure coordinates, an anomalously weak vortex is associated with increased ozone volume mixing ratios throughout the stratosphere, showing local maxima just above the polar tropopause, in the lower-to-middle stratosphere and near the stratopause. Using potential temperature as the vertical coordinate, increased ozone is only present in lower altitudes roughly below 900 K and above about 450 K, even showing weakly decreased values in the lowermost stratosphere beneath for ERAI data. We rationalize these disparate ozone variations by a combination of variability in wave-driven quasi-horizontal mixing and vertical advection by the residual circulation. In particular,

1.  wave-driven anomalous adiabatic up- and downwelling in the polar vortex region cause enhanced polar ozone around 1 hPa and 30 hPa, as well as less ozone equatorwards,

2.  increased downwelling associated with an anomalously weak polar vortex includes a large adiabatic component and acts to simultaneously shift iso-surfaces of potential temperature and ozone also in the lowermost stratosphere, which furthermore results in a lowered tropopause and significantly increased ozone concentrations there, and

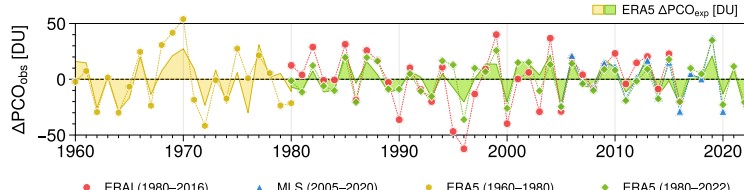

**Figure 9.** Observed polar-cap averaged partial column ozone anomalies $\Delta PCO_{obs}$ per DJF winter season (line plots) for ERAI, MLS and ERA5 data and, for comparison, explained partial column ozone $\Delta PCO_{exp}$ for ERA5 (area plots). See the text for details on the computation.

**Table 1.** Variances of observed polar-cap averaged partial column ozone, $\Delta PCO_{obs}$ defined in eq. (5), explained by interannual variability of $PCO_{exp}$ from eq. (6) associated with T100 anomalies. This table extends the results presented in Fig. 9. Explained variances are provided as the squared correlation coefficients obtained from least-squares linear regressions, where minimum and maximum values were derived from the corresponding 95 % confidence intervals for the correlation coefficients based on Fisher's $z$-transformation.

| Dataset | Time range | Explained variance [%] | | |
|---|---|---|---|---|
| | | min | $r^2$ | max |
| ERAI | 1980–2016 | 28 | 53 | 72 |
| ERA5 | | 48 | 69 | 83 |
| | 1960–1980 | 21 | 55 | 79 |
| | 1980–2022 | 53 | 71 | 83 |
| | 2005–2020 | 47 | 78 | 92 |
| MLS | | 63 | 86 | 95 |

3. the ozone response signature in the middle stratosphere can be explained by consecutive, counteracting anomalous tendencies associated with diabatic downwelling and quasi-isentropic eddy mixing on daily time scales, with varying chronological order depending on altitude.

Our results consistently show that interannual ozone variations are governed by similar dynamical processes as sub-seasonal ozone variability (see, e. g., Lubis et al., 2017; de la Cámara et al., 2018a, b; Hong and Reichler, 2021; Bahramvash Shams et al., 2022). Moreover, the T100 ozone response pattern clearly reflects the underlying ozone transport anomalies when viewed in the latitude-height plane with a vertically resolved response structure. Similar anomaly signatures are observed during Eastern Pacific El Niño events (Benito-Barca et al., 2022, see, e. g., their Fig. 1g). They have furthermore been obtained from modern CMIP6 climate projections based on various climate forcing scenarios (Match and Gerber, 2022, e. g., their Fig. 1).

Finally, referring back to our initial motivation, we found clear similarities between polar vortex–ozone co-variability and
year-by-year variability of zonal mean ozone in general, as measured by the local sample standard deviation provided in Fig. 1.
We may ask to what extent this allows us to constrain polar ozone anomalies during northern hemispheric winters. To do so, we
consider seasonal-mean ozone for DJF from ERAI, MLS and ERA5, each interpolated on equidistant pressure levels between
1 hPa and 300 hPa with 1 hPa vertical resolution and adjusted for linear trends across the corresponding time intervals. The
observed variability of polar-cap averaged partial column ozone (PCO) then reads

$$\Delta\mathrm{PCO}_{\mathrm{obs}}(t) = \left\langle \int_{1\,\mathrm{hPa}}^{300\,\mathrm{hPa}} \delta\overline{\chi}(t,p,\phi)\,\mathrm{d}p \right\rangle_{\phi \geq 60°\mathrm{N}} . \tag{5}$$

Based on the associated T100 regression map for ozone in the latitude-pressure plane, we reconstruct the variations that are
explained by co-variability with the polar vortex,

$$\Delta\mathrm{PCO}_{\mathrm{exp}}(t) = \overline{T}_{100}(t) \cdot \left\langle \int_{1\,\mathrm{hPa}}^{300\,\mathrm{hPa}} b(p,\phi)\,\mathrm{d}p \right\rangle_{\phi \geq 60°\mathrm{N}} , \tag{6}$$

corresponding to the general T100 index time series modulated by a constant prefactor that depends on the individual model
realisation. Finally, the correlation between these two PCO anomaly time series provides a measure for the variance of $\Delta\mathrm{PCO}_{\mathrm{obs}}$
that is associated with seasonal polar vortex strength anomalies. The results obtained for ERAI, MLS and ERA5 are shown in
Fig. 9 and Table 1, which are based on the extended T100 index from ERA5. For a comprehensive analysis we added separate
computations for pre-satellite ERA5 data covering DJF 1960–1980.

Table 1 suggests that around 86 % of polar-cap averaged PCO variations from MLS are related with T100 co-variability.
This significantly differs from ERAI, where only slightly more than half of the variability can be attributed. Explained vari-
ances for ERA5 starting from DJF 1980 turn out to be substantially higher compared with ERAI and furthermore approach
the performance of MLS. The low value for the pre-satellite era in ERA5 suggests that more recent years in reanalyses are
much better constrained by observational data. However, it is unclear whether the remaining differences only result from the
individual model implementations, assimilation schemes and the quality of the measurements. Instead, e. g., given the growing
explained variances for ERA5 with time, the findings may also suggest a much more fundamental change in the interactions
between ozone and large-scale atmospheric dynamics. For example, Calvo et al. (2015) showed that stratospheric zonal wind
and temperature anomalies during Arctic ozone extremes occurred mainly in recent decades due to substantial anthropogenic
ozone depleting substances, indicating that ozone chemistry became increasingly important in governing climate variability.

To sum up, we showed that most of the interannual anomalies of polar column ozone in recent years can be attributed to wave-
driven anomalous dynamics associated with the varying strength of the polar vortex. The knowledge on the mechanisms that
constrain these leading modes of intrinsic ozone variability may help to understand long-term trends in response to external
forcings, i. e., due to evolving concentrations of ozone depleting substances or increased greenhouse gas emissions. (e. g.,
SPARC/IO3C/GAW, 2019) Moreover, feedbacks between anthropogenic climate change and stratospheric dynamics may cause
modifications to these modes of variability. Exploring the extent to which ozone itself is involved here would be worth a closer
look.

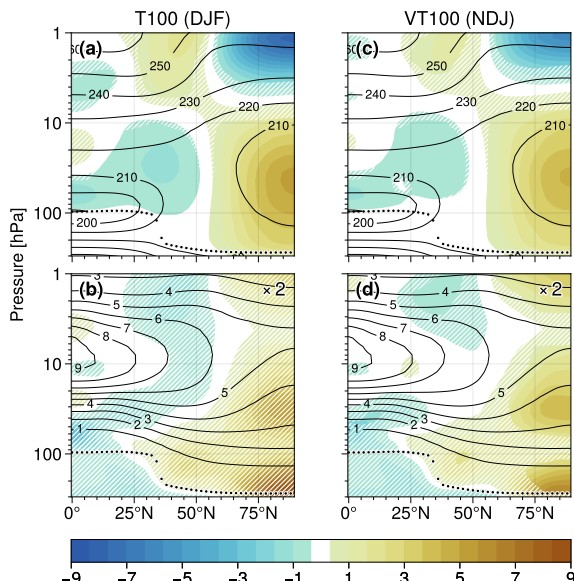

**Figure A1.** Interannual anomalies (seasonal means, DJF) of zonal mean temperature (in units of K, top row) and ozone (bottom row), regressed on **(a, b)** the T100 index (DJF) and **(c, d)** the VT100 predictor (NDJ), which both are described in Sect. 2. Note the time lag for the VT100 time series. The ozone responses (in %) are provided relative to the DJF climatology and have been scaled by a factor 0.5 before plotting. Black contour lines show the corresponding climatology (in K and ppmv for temperature and ozone, respectively) and hatches indicate where the regressions are not statistically significant. The thick dotted lines indicate the seasonal-mean thermal tropopause. All results were obtained from ERA-Interim reanalysis data.

*Data availability.* ERA-Interim data were provided by ECMWF through NCAR. They can be downloaded also from https://apps.ecmwf.int/datasets/data/interim-full-daily. Aura MLS version 5 data are available at https://mls.jpl.nasa.gov. Hersbach et al. (2019) was downloaded from the Copernicus Climate Change Service (C3S) Climate Data Store. Our study contains modified Copernicus Climate Change Service information 2023. Neither the European Commission nor ECMWF is responsible for any use that may be made of the Copernicus information or data it contains.

## Appendix A: T100 and VT100 predictors for linear regressions

In this study, we used anomaly time series based on polar-cap averaged zonal mean temperature at 100 hPa ("T100 index") as the predictors for our linear regression analysis. In Sect. 2, we demonstrated that this index is a suitable measure for the strength of the stratospheric polar vortex. Previous literature on interannual ozone variability often focused on the zonal mean meridional eddy heat flux $\overline{v'T'}$ at 100 hPa ("VT100 index") to quantify the total available wave driving of the stratosphere. For comparison, Fig. A1 provides additional regression maps for seasonal-mean temperature and ozone from ERA-Interim reanalysis data (December–February, DJF), regressed on both the T100 (DJF) and VT100 (November–January, NDJ) predictors. More details and a discussion of the results are included in Sect. 2.

*Author contributions.* FH performed the analyses, supervised by HG and TB. TB initiated the project and processed the ERA-Interim reanalyses. FP provided the MLS observational data. FH wrote the first draft of the paper. All authors contributed in interpreting the results and improving the manuscript.

*Competing interests.* The authors declare that they have no conflict of interest.

*Acknowledgements.* We acknowledge valuable suggestions by two anonymous reviewers that significantly helped to improve this paper. We thank Y. Desille for research on polar vortex–ozone co-variability from modern reanalyses and H. von Heydebrand for extending these studies to chemistry–climate models. We further thank the MLS team for providing the ozone satellite observation data. FH appreciates helpful suggestions by E. Gerber. This work was funded by the Deutsche Forschungsgemeinschaft (DFG, German Research Foundation) – TRR 301 – Project-ID 428312742: "The tropopause region in a changing atmosphere". Throughout this study, we used color codings based on ColorBrewer 2.0 (Brewer, 2022) and Scientific Colour Maps (Crameri et al., 2020; Crameri, 2022). We appreciate the features provided by xarray (Hoyer and Hamman, 2017; Hoyer et al., 2022) and ProPlot (Davis, 2022).

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
