# Peer review of "On the pattern of interannual polar vortex–ozone co-variability during northern hemispheric winter"

_Atmospheric Chemistry and Physics, 2023_

## Author Response (AR1)

**Author response**

June 29, 2023

**On the pattern of interannual polar vortex–ozone co-variability during northern hemispheric winter**

F. Harzer, H. Garny, F. Ploeger, H. Bönisch, P. Hoor, and T. Birner

We thank the two anonymous referees for their careful reading of our manuscript and for their comprehensive feedback. In the following, we list the changes that have been made addressing the remarks of the reviewers. Here, the referees' comments together with our replies from the author comment of May 17, 2023 are printed with blue color. The details on the changes that have been made in the revised version of the manuscript are added beneath. Line numbers in our response refer to the document providing the tracked changes.

**Response to Referee #1**

*Major comments:*

*1. Why the paper only focuses on the northern hemispheric winter season? In the introduction, the authors only mentioned one sentence in line 26. Maybe the authors could provide more information and include some citations to justify their focus on the northern hemispheric winter.*

*2. It is related to question 1. Can the results presented in this paper apply to the southern hemisphere or other seasons? Those other results could be put in the supplementary or the next step. Also, supposed the paper only presents the results on the northern hemispheric winter, it may be a good idea to add "during northern hemispheric winter" in the title to narrow down the research area.*

The polar vortex features substantially stronger dynamical variability in the northern compared to southern hemisphere, which allows for stronger coupling to ozone transport there. A brief check on southern hemispheric polar vortex–ozone co-variability shows significant correlations especially during austral spring (September–November), i. e. the season when polar vortex variability is strongest. This is definitely worth a closer look and more detailed analyses remain for future work. At this point, limiting the scope of our study to the Northern Hemisphere, we will follow the referee's suggestion and adapt the manuscript's title accordingly. Furthermore, a short explanation on this point will be added to the introduction of the manuscript.

The following paragraph has been inserted in lines 58–59:
We limit our analyses to the Northern Hemisphere, since the polar vortex features substantially stronger dynamical variability compared to the Southern Hemisphere. This allows for stronger coupling to ozone transport there.
Furthermore, we adapted the title of the manuscript:
On the pattern of interannual polar vortex–ozone co-variability during northern hemispheric winter

*3. In Figure 2, as expected, the pattern of the anomaly of TEM vertical velocity is similar to the diabatic heating in most regions. But it is interesting to note that large differences exist in regions above 800 K (or 10 hPa) and poleward of 60N. Any explanations here? Also, the diabatic heating patterns are much closer to the temperature patterns in the regions above 800 K (or 10 hPa) and poleward of 60N, compared to the TEM vertical velocity.*

We agree that the results for the TEM vertical velocity in that region may not be fully consistent with the variability patterns that were found for temperature and diabatic heating. We will include more discussion in the text along the lines of the following considerations to address this issue:

- Since parameterizations in ERA-Interim are based on climatological ozone, artificial increments may have been introduced while assimilating temperature in the upper stratosphere. This effect becomes relevant at the polar stratopause presumably due to the large ozone variability (compare, e. g., Fig. 3a) and short radiation time scales there.
- Some additional uncertainties may occur due to the fact that the wind fields in reanalyses are mainly deduced from temperature constraints. Thus, deviations from the real wind distribution are expected especially where the geostrophic approximation is not sufficiently fulfilled, i. e. for the ageostrophic zonal mean residual circulation that is linked to the TEM vertical velocity. Furthermore, mass conservation is not ensured in reanalyses due to data assimilation, which may lead to additional inconsistencies.

The caption of Fig. 3 (labeled Fig. 2 in the previous version of the manuscript) now contains an additional sentence that addresses this issue:

Data assimilation and model approximations may cause inconsistencies in the results, which is the case, e. g., for the TEM vertical velocity in (c) below the polar stratopause compared to the responses of temperature and diabatic heating in panels (b) and (f), respectively.
* * *
4. *The authors focus on diabatic vertical transport and horizontal eddy mixing to explain the anomalous ozone tendency. What about the role of changes in tropopause heights? Figure 4 and Figure 9c in Wang et al. (2020) showed that lower tropopause in cold climates (i.e., last glacial maximum) would lead to increases in ozone concentrations near the tropopause. The strength of the polar vortex may impact the tropopause heights and thus the ozone anomaly. This could be one of the reasons why the high ozone anomaly is gone in isentropic coordinates.*
* * *
We thank the referee for mentioning this additional reference. Indeed, the authors of that study show that extratropical tropopause variability is significantly correlated with stratospheric total column ozone. We discussed this effect for vertically resolved ozone in the lower polar stratosphere, associated with wave-induced T100 variability, e. g., in lines 143–148 in our manuscript and in Figure 5, based on the results for both pressure and potential temperature coordinates. In the revised manuscript we will make sure that this point comes across more clearly.

We have slightly modified the corresponding paragraph in lines 192–199, stating that large relative anomalies occur at the polar tropopause when compared to substantially lower ozone amounts in the upper troposphere.
* * *
*Minor comments:*

1. *In line 115, what about the stronger downward transport due to a stronger BDC (or stronger diabatic cooling) over the polar cap, as shown in Figure 2?*
* * *
We will add a reference in line 115 to Figure 2 in order to strengthen this argument on the relevance of the BDC, accounting for associated anomalously strong poleward and downward transport of ozone.

We inserted a sentence in lines 161–163, providing a link to the stronger residual circulation that is observed in the lower stratosphere in Fig. 3(f) (labeled Fig. 2(f) in the previous version of the manuscript):

This is consistent with the diabatic heating response in Fig. 3(f) that suggests anomalous upwelling (downwelling) in lower (higher) latitudes for ERAI and, hence, a stronger stratospheric residual circulation.

2. *Figure 3 shows clearly that the results remain quite similar on different datasets. It could be helpful to add a figure to plot a similar figure as Figure 3 but on the isentropic levels.*

We agree that a comparison of the datasets on isentropic levels could provide additional evidence for the robustness of the T100 ozone response signature. Up to now, due to substantial computational effort and given the quite consistent results for all three datasets in the pressure coordinate framework in Fig. 3, we have not yet performed these analyses on isentropes for ERA5 and MLS. However, our ongoing research focuses on comparing the ERA-Interim and ERA5 reanalyses, with similar diagnostics derived from high-resolved model level output. The results will be reported elsewhere.

**Response to Referee #2**

*The paper is carefully written and the figures are clear. However, the goal of the paper is not very clear. In some parts, it seems that the main goal is to show that the T100 index is useful to understand polar ozone variability, but then the paper lacks of additional analysis to compare to existing metrics of stratospheric variability (such as eddy heat fluxes) and show the additional value of T100. In other parts, much of the analysis and discussion on the dynamical drivers of polar ozone variability does not seem to add much to the existing knowledge. I therefore suggest performing additional analyses and strengthening the arguments for what is new in this work and why it is important before meriting publication. I have recommended major revision, more detailed comments are listed below.*

We appreciate the comment that the goal of the paper wasn't very clear. We will revise the introduction and text at other places to more clearly reflect the following:

– One motivation for our study is the striking tripole structure of interannual ozone variability over the polar cap, showing isolated variance maxima near the tropopause, in the middle stratosphere and near the stratopause. To reflect this motivation we will move panels (a)–(c) of Fig. 8 to the introduction. As far as we know this particular variability structure has received only little attention in the literature, despite the fact that a large body of work on ozone variations already exists (although most of it focusing on column ozone). We also plan to adapt the title of the manuscript in order to strengthen the link with the vertically resolved patterns of polar vortex–ozone co-variability that are studied in this work. Together with the changes proposed by referee #1, the new title may read "On the pattern of interannual polar vortex–ozone co-variability during northern hemispheric winter".

– Although the dominant roles of vertical advection and horizontal mixing for ozone are well established, our impression is that their detailed interplay with different relative roles at different vertical levels and regions has not been studied as much, e. g. in the lowermost stratosphere.

– Previous work was often performed in either pressure/height or isentropic coordinates. We feel that by combining both of these coordinate frameworks, our analyses provide new insights into how transport variations produce ozone anomalies at different levels and regions, including the relative contributions linked to diabatic and adiabatic downwelling in the polar lower stratosphere, respectively.

For the discussion on the T100 temperature index, please see our reply further below.

We have made several changes to more precisely motivate our study:

– We moved Fig. 8(a)–(c) to the introduction (now labeled Fig. 1). The general pattern of interannual ozone variability motivates our subsequent work on polar vortex co-variability and is discussed in lines 49–56.

- We refer to earlier studies on interannual ozone variability and outline the goal of our work accordingly in lines 48–49:

  While previous studies primarily focused on variations in column ozone, in this work we draw attention to the vertically resolved pattern of year-by-year ozone variability.

- The new title emphasizes our focus on vertically resolved variability structures:

  On the pattern of interannual polar vortex–ozone co-variability during northern hemispheric winter

- Section 3 now includes a more detailed explanation on the goal of this part of our study (lines 149–151):

  In this section, we aim to document this T100 ozone response pattern and provide a comparison based on two different perspectives, i.e. using pressure and potential temperature as the vertical coordinate, respectively.

- In Sect. 5, the summary in lines 282–307 now more clearly refers to the goals of our study, highlighting the differences in the results obtained for pressure and potential temperature as the vertical coordinate, respectively, and how these can be understood from the involved transport processes.
* * *
*The introduction is very general, and previous results on the topic are only briefly discussed in one short paragraph. I would suggest reviewing in more detail the established knowledge of the interannual variability of polar ozone in connection with extratropical stratospheric dynamics (see some suggested references in my general comment above).*
* * *
We will revise the introduction to more specifically address the motivation and goals of our study. Note that some discussion of relevant extratropical dynamics is already included in Section 2.

In lines 34–46, we now mention the literature on interannual ozone variability suggested by the referee.
* * *
*Lines 169-179 (Fig. 6). It is stated in the text that the contributions to the winter-mean ozone tendency of vertical mean advection and horizontal eddy mixing are about the same but with opposite signs, and hence the ozone tendency vanishes in the winter mean "as expected". I have a few comments on this.*

*The first one is that the profiles in Fig. 6f do not show a compensation between both forcings, the negative tendencies from eddy mixing look larger than (at some isentropic levels, almost twice as large as) the positive tendencies from vertical advection. If indeed these are the main forcing terms, that would lead to negative ozone tendencies in the seasonal mean. It would be useful to add the seasonal mean profiles of both the ozone tendency and the ozone tendency derived from the tracer continuity equation to Fig. 6f. How do the winter-mean ozone tendencies in erai compare with the other datasets (particularly with observations)?*

*A second comment is that we should expect an accumulation of ozone in the lower stratospheric polar latitudes throughout the winter due to advection by the residual circulation. This accumulation is mainly driven by downward advection over the poles, which is larger than horizontal eddy mixing in the seasonal average (see e.g., Fig. 3 in De La Camara et al., 2018 for results with a chemistry climate model). This accumulation (late-minus-early winter ozone concentration) has been shown to correlate well with the winter-mean eddy heat flux at 100 hPa (e.g., Weber et al., 2011; Strahan et al., 2016; see also an update of Weber et al. 2011: Fig. 4-13 in Chipperfield and Santee, 2022). Based on this, I was expecting the regression on the T100 index to show larger ozone accumulation (i.e. winter mean positive ozone tendencies, and hence advection larger than eddy mixing) linked to warmer winters at 100 hPa. But Fig. 6f shows the opposite, at least in terms of the unbalance between vertical advection and eddy mixing. I am unsure of what the interpretation is here.*

*Again, plotting the mean ozone tendency from the different datasets regressed on the T100 index may help elucidate this issue.*

We thank the referee for these detailed remarks on this part of our study. Indeed, a closer analysis on the seasonal-mean ozone tendencies, derived from daily ERA-Interim ozone anomalies during December–February 1979–2016, provides evidence for anomalously strong ozone accumulation in the lower stratosphere during years where the polar vortex is weak, which is indicated by a positive T100 index (see figure in the appendix of this author comment). We agree that this is not properly reproduced by the net ozone tendency that takes into account vertical advection and horizontal eddy mixing as the two leading mechanisms of ozone transport. Looking for an explanation, we performed similar computations for the two remaining transport terms in the ozone budget equation (due to horizontal mean advection and vertical eddy mixing), which however confirmed that these contributions are small compared to vertical advection and horizontal eddy mixing (compare the profiles of polar-cap averaged T100 co-variability in the appendix). Furthermore, as discussed in the manuscript, ozone chemistry very unlikely plays a dominant role in the lower stratosphere. Instead, these disparate findings appear to occur due to

- data assimilation of the model output in ERA-Interim,
- uncertainties on the underlying dynamics, i.e. due to less constrained diabatic heating rates, which are derived from the model forecasts only,
- assumptions on the interactions with ozone and the associated feedbacks.

From this we do not expect the ozone budget to be closed in the ERA-Interim reanalyses. The cumulative nature and the smaller magnitudes of the winter-mean tendencies may be even more challenging in that perspective. As pointed out in the manuscript, the residual ozone tendency, obtained from the terms ②+③ in the ozone budget equation, is negative and of non-negligible magnitude, although statistically not very robust (statistical significance is sensitive, e.g. to the latitude range that is chosen for polar-cap averaging; we added another figure in the appendix). We therefore think that such uncertainties only allow for a rather qualitative estimate of the different ozone transport processes. In the corrected version of the manuscript, we will include a similar discussion on the interpretation of our findings. We also hope to shed light on this through more detailed analyses based on ERA5 model output, where we plan to report first results separately in the near future.

In the revised version of the manuscript, the corresponding paragraphs in lines 220–253 now address the uncertainties that are expected from the ERA-Interim reanalyses. Although some qualitative estimates on the different transport processes may be possible, we think that at this point no conclusions can be made regarding absolute budget contributions, motivating our subsequent analyses of sub-seasonal ozone variations provided at the end of Sect. 4.

*Related to this: If a goal of this paper is to show that the T100 index is valuable to understand polar ozone interannual variability, perhaps it'd be interesting to perform a thorough comparison between T100 and an index based on the eddy heat fluxes at 100 hPa (v'T'100), since the latter has been already proven to correlate well with ozone variability (e.g., Chipperfield and Santee, 2022). One way to do it would be to update Fig. 8 and Table 1 with a similar analysis based on v'T'100. Also, it could be useful to analyze not only the interannual variability of winter-mean ozone, but also of winter-mean ozone tendency, since the latter would more strongly depend on the dynamical evolution of the polar vortex and not that much on the initial ozone concentrations at the start of the season.*

Choosing the T100 temperature index for our work is based on the idea to keep this metric as simple as possible and to use a diagnostic based on data that are well constrained in the reanalyses. While this index

is not new, it features a rather direct measure of polar vortex strength. The almost equivalent T100 time series obtained from ERA5 and other datasets confirm the robustness of this index as mentioned in the manuscript. VT100, on the other hand, represents a more involved index, which may be less robust across different data products. Furthermore, although VT100 serves as a proxy for the total available wave driving of the stratosphere above 100 hPa, it is unclear how this wave driving manifests in terms of its latitude-height structure. Having said that, we did check ozone co-variability with VT100 (see figure below), which shows that clear similarities exist in the ozone response patterns obtained for the T100 DJF (Dec–Feb) and the VT100 NDJ time series (Nov–Jan). This observation consistently supports the interpretation on anomalous vertical eddy heat fluxes that precede wave-induced deceleration of the polar vortex by several weeks. (Newman et al., 2001; Polvani and Waugh, 2004)

A new paragraph in lines 133–142 in Sect. 2 now provides a note on using the T100 metric instead of the meridional eddy heat flux for measuring variations of polar vortex strength. Furthermore, the corresponding figure from the author comment of May 17, 2023 has been included in Appendix A of the manuscript.
* * *
*The analysis derived from Fig. 7 is consistent with previous results showing that the initial changes of ozone in the lowermost stratosphere during SSWs are related to eddy fluxes, followed by a balance between vertical advection and mixing (De La Camara et al., 2018; Hong and Reichler, 2021).*
* * *
Thank you, we will add a comment in the manuscript pointing out the similarity with SSW dynamics. In Figure 7, we investigate the main dynamical drivers of ozone transport associated with general sub-seasonal polar vortex variability, which includes more than SSWs.

We inserted a comment in lines 262–265 that points out the similarity of the results with earlier studies on SSW dynamics:

They are consistent with ozone transport taking place, e.g., during sub-seasonal sudden stratospheric warming (SSW) events (de la Cámara et al., 2018a, b; Hong and Reichler, 2021), though here we present lag regressions for the more general case of daily T100 temperature anomalies, which allows for improved statistics and reduced sensitivity, e.g., on SSW definition thresholds.
* * *
*One of the conclusions is that "interannual ozone variations are governed by similar dynamical processes as sub-seasonal ozone variability" (lines 218-222), referring to vertical advection and horizontal eddy mixing. However, these processes are already known to control polar ozone interannual variability (see e.g., Fusco and Salby, 1999; Garcia and Hartmann, 1980; Garcia and Solomon, 1983; Hartmann and Garcia, 1979; Leovy et al., 1985; Ma et al., 2004; Plumb, 2002; Randel, 1993; 1994; 2002). The authors should strengthen their case for what is new in their study.*
* * *
Thanks, we agree that this sentence is too generic. We will reformulate this part of the discussion, more clearly stating where our results confirm previous knowledge and where they add something new (latter primarily related to the latitude-height structure of ozone variations and what can be learned by comparing pressure and isentropic coordinates).

In lines 301–305, we now point out the consistency of our results with previous work and refer back to the vertically resolved structure of the ozone response:

Our results consistently show that interannual ozone variations are governed by similar dynamical processes as sub-seasonal ozone variability (see, e.g., Lubis et al., 2017; de la Cámara et al., 2018a, b; Hong and Reichler, 2021; Bahramvash Shams et al., 2022). Moreover, the T100 ozone response pattern clearly reflects the underlying ozone transport anomalies when viewed in the latitude-height plane with a vertically resolved response structure.